# Influence of Hydrophobicity of Backbone Polymer in Thermo-Responsive Hydrogel with Immobilized Amine on Cycle Capacity for Absorption and Recovery of CO_2_

**DOI:** 10.3390/polym11061024

**Published:** 2019-06-10

**Authors:** Yuma Nagasawa, Yoshimi Seida, Takehiko Gotoh, Eiji Furuya

**Affiliations:** 1Department of Applied Chemistry, Meiji University, Kawasaki 214−8571, Japan; n7y2u6@gmail.com (Y.N.); egfuruya@meiji.ac.jp (E.F.); 2Natural Science Laboratory, Toyo University, Tokyo 112−8606, Japan; 3Department of Chemical Engineering, Graduate School of Engineering, Hiroshima University, Higashi Hiroshima 739−8527, Japan; tgoto@hiroshima-u.ac.jp

**Keywords:** thermo-responsive hydrogel, tertiary amine, CO_2_, adsorption, equilibrium sorption model

## Abstract

The chemisorption process with amines is the major separation and recovery method of CO_2_ because of its high processing capacity and simplicity. However, large energy consumption for the desorption of CO_2_ is also associated with the process. To develop a separation and recovery process that is capable of desorbing CO_2_ at low temperatures and with minimal energy consumption, polymer hydrogels with a lower critical solution temperature (LCST) polymer network and amine groups immobilized in the polymer network of the hydrogels were exploited. Thermo-responsive amine gels with a series of hydrophobicity of polymer networks were systematically synthesized, and the influence of the hydrophobicity of the gels on the CO_2_ desorption temperature and cycle capacity (CO_2_ amount that can be separated and recovered by 1 cycle of temperature swing operation) was investigated using slurries with the series of gels. A significant decrease in the CO_2_ desorption temperature and increase in the cycle capacity occurred simultaneously by lowering the LCST of the gels via hydrophobisation of the polymer network. Based on an equilibrium adsorption model representing the CO_2_ separation and a recovery system with the gel slurries, an analysis of the system dynamics was performed in order to understand the recovery mechanism in the process.

## 1. Introduction

Chemisorption is the major practical method for post-combustion CO_2_ recovery due to its process simplicity and treatment capacity. Recently, chemisorption has been applied to the capture and removal of CO_2_ from the atmosphere (Direct Air Capture: DAC [1,2,3]), as have other conventional separation methods. Amine-based chemisorption has been applied practically in post-combustion CO_2_ recovery; however, the large amount of energy consumption that is associated with the regeneration/desorption treatment, which occurs at high temperatures (>393 K), is a drawback of the process [4]. Tertiary amines and hindered amines have shown promise as CO_2_ adsorbents; with these adsorbents, energy consumption and process temperature can be significantly reduced. On the premise that waste heat can be utilized, it is required that the chemisorption-based CO_2_ separation process be driven at temperatures provided by low-grade heat sources. The desorption of CO_2_ with tertiary amines is less energy-consuming compared to those with other amines. This is because tertiary amines adsorb dissolved CO_2_ as bicarbonate ions via electrostatic interactions (r1), without undergoing reactions involving covalent bonding, such as carbamate formation, which occurs for 1st and 2nd amines.
(r1)$$-R_{1}R_{2}N+H_{2}O+{\mathrm{CO}}_{2}⇄-R_{1}R_{2}{\mathrm{NH}}^{+}+{\mathrm{HCO}}_{3}^{-}$$

In the present study, thermo-responsive hydrogels with tertiary amines were used as energy-saving adsorbents for the chemisorption-based CO_2_ separation and recovery process. Tertiary amines, immobilized in cross-linked hydrogels, exhibit temperature-responsiveness of their p*K*_a_ in response to the swelling/collapsing of the gels, which is the key property of the gels for the adsorption of CO_2_ at the low desorption and regeneration temperature [5,6,7]. In this separation and recovery system of CO_2_, the adsorption and desorption of CO_2_ onto the gels are operated by controlling the swelling of the gels via temperature swing. The influence of the polymer characteristics on the p*K*_a_ of the amine groups in the gels has been studied in terms of the hydrophobicity of the polymers by neutralization titration [8], pH change in aqueous solution in response to the phase change in the thermo-responsive polymers [9], and temperature dependence of the swelling volume and pH of the gels [10]. The influence of the adsorption of CO_2_ on the lower critical solution temperature (LCST) of the polymer in the amine gels has also been reported [11]. In the gel system, CO_2_ gas is adsorbed on protonated amines in the gels as bicarbonate ions, when the pH of the gel system is below the p*K*_a_ of the amines, at a low temperature below the LCST of the gel (the temperature at which the gel swells). Conversely, in principle, the bicarbonate desorbs and is released as CO_2_ gas when the pH of the gel system is above the p*K*_a_ of the amines at a high temperature above the LCST [5,6]. Therefore, the LCST of the polymer used as the backbone of the gels (polymer networks) is the key index in the CO_2_ separation and recovery process, i.e., the CO_2_ adsorbed on the tertiary amines in the gels, at a low temperature, is released at a high temperature after the LCST of the gels is attained. The temperature-dependent phase behaviour of the thermo-responsive gels affects the p*K*_a_ of the amines in the gels, and the p*K*_a_ change in the amines is most pronounced after the LCST. Therefore, the performance of the CO_2_ separation and the recovery process depends heavily on the LCST. We focused on controlling the LCSTs of the gels for the development of an effective, simple, energy-saving CO_2_ recovery system. The goal of this study is the development of a gel system which reduces the desorption temperature of CO_2_ without affecting the CO_2_ adsorption capacity of the thermo-responsive gels with LCST polymer network. The factors available for controlling the LCST in the thermo-responsive polymer gels are: the composition of the backbone polymers [12], ionic group contents [13], control of ionic groups ionisations [14], solvent [15], molecules [16], and additives [17], according to the Flory-Huggins-Tanaka’s theory [18]. In the development of CO_2_ chemisorption-based separation and recovery process, it is important to evaluate the performance of the separation with a slurry system because of its process simplicity, adaptability to existing infrastructure and processing capacity. The key process is a gas-liquid reaction in the slurry system, and the efficiency of the gas-liquid reaction depends on the interface area between the gas and liquid, which can be improved by various configurations of equipment (design of contactors) and operations of the processes in the slurry system. In a previous study, the influence of the solvent composition and NaCl, which reduce the LCST of the thermo-responsive gels, on the desorption temperature and the cyclic capacity for CO_2_ was investigated using a slurry system with a well-crushed, cross-linked thermo-responsive N-isopropylacrylamide-N-3-dimethylaminopropylacrylamide copolymer hydrogel as the CO_2_ adsorbent. Alcohol is known to absorb CO_2_ better than water, and the CO_2_ solubility decreases in higher alcohols [19,20,21,22]. Adding 10%–30% of ethanol, 2-propanol, 2-butanol, *tert*-butanol, and dimethylsulfoxide to the slurry reduced the desorption temperature of CO_2_, as was intended; however, the addition caused a significant reduction in the cycle capacity for CO_2_ [23]. The addition of NaCl resulted in a decrease in both CO_2_ solubility and the adsorption capacity of the gel with the increase of its concentration (0.01–0.1 M) in the slurry [23]. 

Hydration becomes difficult when the polymer becomes hydrophobic due to the LCST events in aqueous systems. The ionisation of the amines immobilized in the polymer network increases the hydrophilicity of the polymer and works to swell (stretch) the polymer chains. The hydrophobic polymer chains require more protons than the hydrophilic polymer chains to ionise the amines required to swell the hydrophobicised polymer chains. This means that the p*K*_a_ of the immobilized amines becomes relatively small in the hydrophobic polymer chains, since the dissociation equilibrium of the amine groups shifted to a relatively low pH region in the polymer with the hydrophobic chains [10]. Therefore, in the polymers with amine groups, which exhibit a large hydrophobic-hydrophilic change in response to the temperature change, significant p*K*_a_ change will be induced, which results in effective adsorption/desorption of the gels for CO_2_.

In this study, thermo-responsive amine gels with series of hydrophobicity of polymer networks were prepared using N-isopropylacrylamide (NIPA, LCST of polymer = 306 K in aqueous system), N-*tert*-butylacrylamide (NTBA, LCST < 273 K), and N-3-dimethylamino-propylacrylamide (tertiary amine). The temperature swing adsorption/desorption tests for CO_2_ were performed using slurries with the series of gels to understand the effect of the hydrophobisation of the polymer network on both the desorption behaviour of CO_2_ and the cyclic capacity of the gels for CO_2_. The adsorption/desorption operation was performed by repeating the temperature swing of the gel slurries under an atmosphere of CO_2_-N_2_ gas. An equilibrium adsorption model for the gel slurry system was developed based on the phenomenological observations occurring during the CO_2_ adsorption/ desorption process to understand the mechanism of the CO_2_ recovery in the gel system. The adsorption/desorption mechanism and the cycle capacity of this process were examined based on the equilibrium adsorption model, which considers the characteristics of the gels.

## 2. Experimental Section

### 2.1. Materials 

N-isopropylacrylamide (NIPA, main monomer) and N-3-dimethylaminopropylacrylamide (DMAPAA, tertiary amine, p*K*_a_ = 10.35) were supplied from KJ Chemicals Corporation, Tokyo, Japan. N-*tert*-butylacrylamide (NTBA, co-monomer) was purchased from Tokyo Chemical Industry Co. Ltd., Tokyo, Japan. *N,N*’-methylenebisacrylamide (BIS, crosslinker), *N,N,N’,N’*-tetramethylenediamine (accelerator), *N,N*-dimethylformamide, and ammonium persulfate (initiator) were purchased from Fujifilm Wako Pure Chemical Co., Osaka, Japan. The monomers were purified before use by conventional methods. The other chemicals were used as received.

### 2.2. Synthesis of the Gels

A series of NIPA-NTBA-DMAPAA copolymer gels, listed in Table 1, were synthesised by free radical polymerisation in a glass vial with *N*,*N*’-methylenebisacrylamide, *N*,*N*,*N*’,*N*’-tetramethylenediamine, and ammonium persulfate in 50–75% DMF, at 333 K. The total monomer concentration was 700 mM. The synthesised gels were crushed into pieces, and then washed with a large amount of distilled water to remove the remaining chemicals. The washed gels were dried in an electric oven at 353 K for several days. The dried gels were crushed and sieved with a 350 μm-mesh sieve, and were used in the following experiments.

### 2.3. IR Measurement

ATR-FTIR spectra of the synthesised gels were recorded on an FTIR spectrometer (Thermo Fisher Scientific, Nicolet™ iS™ 50 FT-IR, Yokohama, Japan) at ambient temperature. The scans were repeated 16 times in the range of 400 and 4000 cm^−1^. The obtained spectra were normalized based on the peak intensity at 1644 cm^−1^. 

### 2.4. Swelling Behaviour of the Gels

The prepared gels were used in the slurry form for the adsorption/desorption test of CO_2_ by the method shown below. The gels adsorb and desorb bicarbonate ions with a change in their swelling volume during the temperature swing in the test. Therefore, the temperature dependence of the swelling of the gels in the aqueous solution, with bicarbonate ion, was investigated using a reliable method developed by Seida et al. [24]. A concentration of bicarbonate ions that was the same in the CO_2_ adsorption/desorption experiment was employed in the observation of the swelling behaviour of the gels. A fraction of each gel (0.1 g) was placed in a glass tube filled with 5 mL of 0.04 M aqueous sodium hydrogen carbonate. The glass tubes were sealed so as to not create a gas space in the sealed tubes. The glass tubes were heated stepwise in a temperature-controlled water bath (AS ONE, Thermax TM-3, Tokyo, Japan), and the swollen volumes of the gels were measured after 60 min of heating at each temperature, from the packing height of the gels in the glass tubes.

### 2.5. Influence of the Gels on the Slurry pH

The influence of the gels on the pH of the slurries was examined, and the results were reflected in the modelling of the CO_2_ adsorption process under study. A fraction (0.1 g) of each dry gel was mixed with 40 mL of distilled water and kept at room temperature for 24 h. The pH values of the slurries were measured using a pH meter (Horiba Ltd., LAQUA twin pH-22B, Kyoto, Japan). Subsequently, the slurries were ground for 30 s using an electric blender (Iwatani, Crush Millser, IFM-C20G, Tokyo, Japan) to obtain slurries of finely crushed gels. The crushed slurries were forced through a 30 µm sieve at room temperature. The pH values of the slurries were measured again (pH after crush). After that, the slurries were heated at 353 K over the LCST of the gels in the slurries for 10 min and were filtered with a 0.2 µm membrane filter. The pH values of the filtrates were measured using the pH meter. All the pH measurements were performed under ambient conditions. 

### 2.6. CO_2_ Adsorption/Desorption Cycle Test

The CO_2_ adsorption/desorption performance of the slurries with the series of the gels was evaluated using a continuous stirred tank reactor with constant gas flow (Figure 1). A fraction (1 g) of each dry gel was introduced into the reactor filled with 400 mL of distilled water, and the slurries were kept overnight under vigorous stirring at room temperature. The CO_2_ gas, balanced with N_2_, was bubbled into the reactor at a flow rate of 7 mL·s^−1^, while the slurries were vigorously stirred. The temperature of the reactor was raised from 293 K to 353 K, at 2 K/min using a PID programmable temperature controller (AS ONE, Thermax, TM-3). The temperature of the reactor was kept at 353 K until the concentration of CO_2_ in the effluent gas from the reactor reached a constant value. After that, the reactor was cooled to 293 K at 1 K/min. This temperature swing operation was repeated several times to confirm the reproducibility of the adsorption/desorption of CO_2_. The heating and cooling rates were determined by a preliminary experiment so as not to rate-limit the adsorption/desorption of CO_2_. The temperature of the slurry was measured directly using a copper-constantan thermocouple. The concentration of CO_2_ in the effluent gas and pH of the slurry were measured with a non-dispersive infrared gas analyser (T&D Co., Matsumoto, Japan, CO_2_ recorder, TR-76Ui-S) and pH meter (DKK-TOA Co., Ltd., Tokyo, Japan, DKK HM-25G), respectively. The water evaporated in the reactor was recovered by a reflux condenser placed at the top of the reactor. The amount of CO_2_ desorbed from the slurries in the heating process was calculated from the desorption time-course of CO_2_ concentration during the heating process. The CO_2_ desorbed in the heating process indicated the amount of CO_2_ that could be adsorbed in one cycle of the temperature swing operation. This cycle capacity of the gel system for CO_2_ under the experimental conditions was employed in this study. 

## 3. Results

### 3.1. FTIR Spectra

The FTIR spectra of the gels are shown in Figure 2. A decrease in the 1171 and 1129 cm^−1^ peak intensities with the decrease of NIPA composition were observed. An increase in the 1224 and 1365 cm^-1^ peak intensities with the increase of NTBA composition in the gels can be attributed to the difference in the composition between NIPA and NTBA in the gels. 

### 3.2. Swelling Behaviour of the Gels

Figure 3a–c indicate the temperature dependence of the swelling of the series of gels. The volume change shifted toward the relatively low-temperature region with the increase in NTBA in the gels (No.1→5, No.6→7). Measurements of the enthalpy changes in the volume phase change of the gels were carried out by means of differential scanning calorimeter using Seiko co. ltd, DSC200, but it was not observable well. Most of the gels became cloudy at the temperatures at which the gels showed collapsed phase. Therefore, the LCST of each gel was defined as the clouding temperature. Since the samples of gels Nos. 5 and 8 were clouded even at low temperatures, the temperatures at the intersection between the lines of the swelling curves at high temperature at which the gels showed collapsed phases and the differential lines of the swelling curves near the collapsed phase were considered as the LCST, being indicated by cross marks on the swelling curves. The LCST of each gel determined in this manner is 320, 313, 310, 306, 292, 332, 322, 309, 326, 313, 323, 338 K, in order of the gel number. The decrease of LCST with the increase of NTBA depending on the DMAPAA content can be confirmed from the LCST plot shown in Figure 3d. The LCST shifted to the low temperature side with the increase of NTBA and shifted to the high temperature side with the increase of DMAPAA. The LCST can be controlled simply by introducing NTBA to the network. Furthermore, the NTBA-DMAPAA copolymer gels (No. 10–12) exhibited temperature responsiveness. The gels with an NTBA rich polymer network exhibited phase change over a broad temperature range (No.4-10). The temperature where the gels collapse decreased with the increase in the hydrophobicity of the gels.

### 3.3. Influence of the Gel on the pH of the Slurry

Figure 4 shows the pH of the slurries before and after the gels are finely crushed in the slurries, and the pH of the filtrates of the finely crushed slurries that were heat-treated at 353 K. The slurries obtained after the fine crush tend to exhibit higher pH values than those of the slurries before the crushing treatment. The pH decreases significantly with the increase in hydrophobicity of the gels (an increase in NTBA content) and increases considerably with the amine content. The pH values of the filtrates obtained by filtration of the phase separated slurries with 0.2 μm membrane filter (i.e. the pH after removing the gel from the 0.2 μm membrane filter) were below the pH values of the slurries before the crushing treatment. Using the slurry, in which the cross-linked gel fine particles are dispersed, the solution is easily shifted towards the alkaline pH, whereby the pH can be alternated within a range effective for the adsorption and desorption of CO_2_. When the gels in the slurries are miniaturized, the pH of the slurries will increase, depending on the size of the gels (Appendix B). Dissolution of CO_2_ from atmosphere into water decreases the pH of water due to proton release via dissociation of carbonate acid in water. The increase in the pH of the slurry is characteristic of the system with the gel particles with a cationic fixed charge. The finely crushed gels increase the pH of the slurry more than the course gels. 

### 3.4. CO_2_ Adsorption/Desorption Behaviour in the Cycle Test

The concentration of CO_2_ in the effluent gas increased gradually with temperature at the first stage of the heating process; then, a large peak due to the desorption from deprotonated amines appeared at a temperature (Figure 5a), which was not observed for pure NIPA gel, as well as the non-gel systems [25]. The large desorption peak appeared at a relatively high temperature after the LCST event of the gels. For the cooling process, a broad convex peak appeared at the low-temperature region, i.e., below 323 K, indicating the adsorption of CO_2_. The pH in the slurries also revealed behaviours that are characteristic of the gel slurry system, as shown in Figure 5b. The pH decreased with temperature in the heating process and increased in the cooling process. The characteristic changes in the CO_2_ concentration profile and pH profile are observed depending on the gels in the adsorption/desorption test, which correspond to the phase behaviour and the LCST event of the gels. 

Figure 6 shows the range of pH change of the slurries during the temperature swing, the temperature of CO_2_ desorption peak, and the desorption amount of CO_2_ (cycle capacity) obtained in the CO_2_ adsorption/desorption cycle test. The desorption amount increased with the increase of NTBA in the gels (No. 1→4, No. 6→7) and the increase in the amine (DMAPAA) content. The introduction of hydrophobicity in the backbone thermo-responsive polymer network resulted in a decrease in the desorption temperature without reducing the cyclic capacity of CO_2_. Conversely, the desorption amount was small in gels Nos. 5 and 8. The gels with low LCST exhibited hydrophobic collapse state at the low temperature, and thus adsorbed less CO_2_. The gels that formed a collapse phase due to hydrophobic polymer networks had a small adsorption capacity for CO_2_, even at low temperatures (No. 5, 8).

## 4. Discussion

For the composition of the gels, the gel with a high amine content with a sharp volume phase transition at a low temperature was shown to be ideal for the effective, energy efficient recovery of CO_2_; however, an increase in the amount of NTBA and DMAPAA resulted in gels exhibiting a gentle temperature-dependent volume change, as indicated in this study.

The desorption peak temperature of CO_2_ correlated with the LCST, and the desorption peak temperature shifted to a low temperature with the increase of NTBA in the gels. No. 5 gel desorbed a very small amount of CO_2_ (small cyclic capacity). No. 5 gel exhibited a collapse phase due to its strong hydrophobicity at low temperature, where CO_2_ adsorbs. The pH of the slurry of gel No. 5 was less than 6 at the low temperature. It was shown that the amines did not work as charged sites. The pH in the gel was not as different from the solution pH (pH_L_) as that of gel No. 5. In view of the pH dependency of the dissociation equilibrium of amine in the gel (*K*_a,g_), most of the amines will be protonated in gel No. 5 at the pH of the slurry. However, the protonated amines may not function effectively, resulting in an extremely small cycle capacity for CO_2_, as shown in Figure 6. Since the pH of the slurry (= pH of the liquid phase of the slurry = pH_L_) is as low as 6 or less in the low-temperature range, the ionic groups are possibly not functioning in the hydrophobised polymer network. It is necessary to analyze the state of the amines. In the water system, the dissolution of CO_2_ results in a weakly acidic water pH due to proton release by dissociation of carbonic acid, as mentioned above. However, the slurries exhibit a weak alkaline pH at low temperature in the presence of the amine gel. The charged (protonated) amine groups in the gels decreased with a shift in p*K*_a_ of the amines at high temperatures, which causes the pH_L_ to shift to acidic, as in the case of water. This pH_L_ shift phenomenon is characteristic in the aqueous system with the amine gels, and will work effectively in the separation of CO_2_ under study, to gain the cycle capacity of the gels for CO_2_. 

### Theoretical Consideration

Herein, the adsorption and recovery characteristics of the gels for CO_2_ were examined based on an equilibrium model of the adsorption behaviour in the gel-slurry system. Various models have been shown for the equilibrium CO_2_ sorption to amines in liquid systems [26,27,28,29]. Equilibrium CO_2_ sorption models have also been reported by Xiao et al. [30] for tertiary amines in liquid systems. The characteristics of the gel-slurry system in this study are the system in which the ionic gel is dispersed and the ionic group changes its equilibrium constant (p*K*_a_), depending on the temperature and pH of the system. This system is expressed by the following model equations. In the model, gas-liquid equilibrium of CO_2_ gas, mass balance in the liquid, and the gel phases in the slurry, dissociation equilibrium of carbonic acid and bicarbonate, electrical neutrality condition, ionic product of water, dissociation equilibrium of amine, and Donnan’s equilibrium at the liquid-gel interface are taken into consideration. Henry’s law was applied to the dissolution equilibrium (gas-liquid equilibrium) of CO_2_. The influence of the vapour pressure of water, which reduces the partial pressure of CO_2_, was considered in the gas-liquid equilibrium. The dissolution of CO_2_ is an exothermic reaction; thus, the solubility of CO_2_ decreases with increasing temperature. The amount of CO_2_ that is in equilibrium with the partial pressure of CO_2_ (*p*_CO2_) in the gas phase was calculated by Equation (1). Henry’s constant at each temperature was determined by Equation (2) in this calculation [27]. The saturated water vapour pressure was considered using the Tetens equation [31] in this calculation.
(1)$$p_{CO2}=Hx_{CO2}\;\mathrm{where}\;x_{CO2}=n_{CO2}/\left(n_{CO2}+n_{H2O}\right)$$
(2)ln(H)=11.181−1949.6/T+0.0822ln(T) (in pure water)where *H* and *T* are the Henry constant and absolute temperature, respectively; *n*
_H2O_ and *n*
_CO2_ represent moles of water and dissolved CO_2_ in the liquid phase, respectively. Most of the dissolved CO_2_ exist in the form of CO_2_ in the water system; however, the total amount of dissolved CO_2_ and H_2_CO_3_(*aq*) defined by H_2_CO_3_* (Equation (3)) was used in the following equilibrium analysis based on the Ishida’s report [27]. The concentration of H_2_CO_3_* is equivalent to *x*_CO2_ from the mass balance.
(r2)$${\mathrm{CO}}_{2}\left(\mathrm{aq}\right)+H_{2}O\left(l\right)⇄H_{2}{\mathrm{CO}}_{3}\left(aq\right),\;Kr_{2}\sim 650\;[29]$$
(3)$$\left[H_{2}{\mathrm{CO}}_{3}{}^{*}\right]=\left[{\mathrm{CO}}_{2}\left(aq\right)\right]+\left[H_{2}{\mathrm{CO}}_{3}\left(aq\right)\right]$$

The dissolved CO_2_ produces carbonic acid in water. The carbonic acid dissociates by Equation (r3) and (r4). The equilibrium constants for each reaction are shown by Equations (4) and (5). The temperature dependent dissociation constants of each dissociation reaction were calculated using Equations (6) and (7), reported by Plummer and Busenberg [28].
(r3)$$H_{2}{\mathrm{CO}}_{3}⇄H^{+}+{\mathrm{HCO}}_{3}^{-}$$
(r4)$${\mathrm{HCO}}_{3}^{-}⇄H^{+}+{\mathrm{CO}}_{3}^{2-}$$
(4)$$K_{a1}=\frac{\left[H^{+}\right]\left[{\mathrm{HCO}}_{3}^{-}\right]}{\left[H_{2}{\mathrm{CO}}_{3}\right]}(\mathrm{degree}\;\mathrm{of}\;\mathrm{dissociation},\;\alpha )$$
(5)$$K_{a2}=\frac{\left[H^{+}\right]\left[{\mathrm{CO}}_{3}^{2-}\right]}{\left[{\mathrm{HCO}}_{3}^{-}\right]}\;(\mathrm{degree}\;\mathrm{of}\;\mathrm{dissociation},\;\beta )$$
(6)$$pK_{a1}=-(-356.3094-0.06091964T+\frac{21834.37}{T}+126.8339\mathrm{Log}\left(T\right)-1684915/T^{2})$$
(7)$$pK_{a2}=-\left(-107.8871-0.0325284T+\frac{5151.79}{T}\;+38.925611\mathrm{Log}\left(T\right)-563713.9/T^{2}\right)$$

Total absorbed CO_2_
(8)$$\left[H_{2}{\mathrm{CO}}_{3}^{*}]+[{\mathrm{HCO}}_{3}^{-}]+[{\mathrm{CO}}_{3}^{2-}\right]=c_{0}$$

Dissociation equilibrium of water
(9)$$K_{w}=[H^{+}]\left[{\mathrm{OH}}^{-}\right]$$

Charge balance (electric neutrality) in the liquid phase in the slurry
(10)$$\left[H^{+}\right]=\left[{\mathrm{HCO}}_{3}^{-}\right]+2\left[{\mathrm{CO}}_{3}^{2-}\right]+\left[{\mathrm{OH}}^{-}\right]$$

From Equations (4)–(10), the concentration of each species in the liquid phase in the slurry is derived as a function of the proton concentration (*c*_H_), as follows.
(11)$$\left[{\mathrm{HCO}}_{3}^{-}\right]=\frac{K_{a1}C_{H}}{c_{H}^{2}+c_{H}K_{a1}+K_{a1}K_{a2}}c_{0}$$
(12)$$\left[{\mathrm{CO}}_{3}^{2-}\right]=\frac{K_{a1}K_{a2}}{c_{H}^{2}+c_{H}K_{a1}+K_{a1}K_{a2}}c_{0}$$
(13)$$\left[H_{2}{\mathrm{CO}}_{3}^{*}\right]=\frac{c_{H}^{2}}{c_{H}^{2}+c_{H}K_{a1}+K_{a1}K_{a2}}c_{0}$$

The amount of CO_2_ absorbed can be obtained as the function of *c*_H_ and *c*_0_. Donnan equilibrium is considered between the liquid and the gel phases in the slurry. The dissociation reaction of the amine groups and its equilibrium constant, *K_a_,*_g_, (degree of dissociation = *β*_g_) are
(r5)$$-{\mathrm{NR}}_{2}H^{+}=-{\mathrm{NR}}_{2}+H^{+}$$
(14)$$K_{a,g}=\frac{\left[-{\mathrm{NR}}_{2}\right]\left[H^{+}\right]}{\left[-{\mathrm{NR}}_{2}H^{+}\right]}=\frac{\left[X\right]\left[H^{+}\right]}{\left[X^{+}\right]}\;\to (\mathrm{degree}\;\mathrm{of}\;\mathrm{dissociation},\;\beta_{g})$$

Mass balance of amine groups
(15)$$\left[X^{+}\right]={\left[X\right]}_{0}-\left[X\right]$$

Donnan equilibrium at the liquid-gel interface
(16)$$c_{i,L}=p^{zi}c_{i,g}$$where *c_i,_*_L_ and *c_i,_*_g_ are the concentrations of *i* ion in the liquid and gel phases in the slurry, respectively. Donnan ratio and the valency of the *i* ion are indicated by *p* and *z_i_*, respectively. The charge balance (electric neutrality) in the gel phase in the slurry is given by the following:(17)$$\left[-{\mathrm{NR}}_{2}H^{+}\right]+\left[H^{+}\right]=\left[{\mathrm{HCO}}_{3}^{-}\right]+2\left[{\mathrm{CO}}_{3}^{2-}\right]+\left[{\mathrm{OH}}^{-}\right]$$

By substituting Equations (11)–(16) into Equation (17) of the charge balance, following Equation (18) is obtained.
(18)$$p^{2}{\left[H^{+}\right]}_{L}{}^{2}+p^{3}{\left[H^{+}\right]}_{L}\left(\left[X_{0}\right]+K_{a,g}\right)-p^{2}\left({\left[H^{+}\right]}_{L}{\left[{\mathrm{HCO}}_{3}^{-}\right]}_{L}+K_{w}\right)-p\left(2{\left[H^{+}\right]}_{L}{\left[{\mathrm{CO}}_{3}^{2-}\right]}_{L}+K_{a,g}{\left[{\mathrm{HCO}}_{3}^{-}\right]}_{L}+\frac{K_{a,g}K_{w}}{{\left[H^{+}\right]}_{L}}\right)-2K_{a,g}{\left[{\mathrm{CO}}_{3}^{2-}\right]}_{L}=0$$

From the experimental result shown in Figure 4, the influence of the cationic charge of amine groups on the pH of the slurry was suggested. Yang et al. reported the pH change in the solution with the thermo-responsive ionic linear polymer [9]. The results in this study indicate the influence of cationic (amine) groups immobilized in the thermo-responsive polymer on the solution pH for the cross-linked gels. The hydroxide ions in the liquid phase in the slurry will be electrostatically attracted to the protonated amine groups on the surface of the gels; consequently, hydroxide ions that exist to neutralize the surface charge affect the pH of the slurry (pH in the liquid phase in the slurry, pH_L_). In the model, the influence of surface charge on the gel was introduced as a surface charge contribution factor *β* = (amount of surface charge affecting the pH in the slurry)/(total amount of charge in the gels). The details of the definition of *β* in the model are shown in the Appendix B. 

Regarding the equilibrium, the absorption amount of CO_2_ can be calculated as a function of the temperature by solving the above-mentioned equations of the system. According to the experimental conditions used in this study, the amount of dissolved CO_2_ [H_2_CO_3_*] that is in equilibrium with CO_2_ in the gas phase, was calculated at first, based on Henry’s law. Subsequently, the composition in the liquid phase in the slurry and pH in the liquid phase (pH_L_) were calculated, in which the mass balance of CO_2_, dissociation equilibrium of carbonate acid and its dissociated ion, and electric charge balance were satisfied. Subsequently, the composition in the gel phase was calculated using the composition in the liquid phase. Based on the Donnan equilibrium between the gel and the liquid phases in the slurry, the composition of the ions in the gel was calculated to satisfy the charge balance in Equation (18) and dissociation equilibrium of carbonate, fixed charge and water, respectively. The total amount of CO_2_ adsorbed in the gel-slurry system (*Q*) is the sum of the amount of H_2_CO_3_^*^, HCO_3_^-^, and CO_3_^2-^ in the slurry. The cycle capacity can be calculated by Equation (19) on the equilibrium model basis. Where *T*_L_ and *T*_H_ are operation low temperature for CO_2_ adsorption and operation high temperature for CO_2_ desorption, respectively.
(19)$$cycle\;capacity=\;Q_{T_{L}}-Q_{T_{H}}$$

Figure 7a shows the temperature dependency of the amount of CO_2_ absorption, *Q* (total amount absorbed CO_2_ in the slurry), and Donnan ratio as functions of p*K*_a,g_. Depending on the p*K*_a,g_ of the amine groups in the gels, the amount of the absorption decreases with increasing temperature. The Donnan ratio tends to increase with the temperature, indicating a decrease in the difference of concentration of ions between the liquid and gel phases. When p*K*_a,g_ decreases (it will occur with the collapse of the gel) at a high-temperature region, the CO_2_ that was adsorbed at the low temperature desorbs at the high-temperature region, depending on the temperature dependence of p*K*_a,g_ (refer the route of the arrow in the figure). The volume change and the p*K*_a_ change of the amine groups in the gel will be linked with each other. The p*K*_a_ affects the number of protonated fixed-charge sites (the number of electrostatic adsorption sites); thus, the *Q* depends on p*K*_a,g_ significantly. Figure 7b shows the dependence of the pH in the gel (pH_g_) and the pH in the liquid phase of the slurry (pH_L_) on the p*K*_a,g_ of the amine groups in the gel, as a function of temperature. For an aqueous solution system without the gels, the solution pH tends to increase with the temperature, as shown by dotted lines in the figure. The pH in the gel (pH_g_) becomes higher than the pH_L_ in the liquid phase due to the Donnan equilibrium, and the pH in the gel increases with increasing temperature, as is the case with the aqueous solution system when the p*K*_a_ of the amine (p*K*_a,g_) is constant. When the p*K*_a,g_ becomes small due to the temperature, the pH difference between the interior and exterior of the gel, and the pH value itself in the system, decrease considerably with the increase in temperature, as was observed experimentally in the slurries under study. The concentrations of ions in the gel phase are higher than in the liquid phase in the slurry due to the Donnan equilibrium (=existence of the fixed charge groups), resulting in the larger change in the proton concentration depending on the temperature and the p*K*_a,g_.

Figure 8a indicates the influence of *β* on the pH inside (pH_L_) and outside gel (pH_g_). The pH in the liquid and gel phases increases with an increase in *β*. The charge density (concentration of amine) in the gel is independent, irrespective *β*. Thus, the pH_g_ is independent to *β* in this model. Furthermore, the amount of absorbed CO_2_ depends on *β*, as shown in Figure 8b. The influence of *β* on the absorption amount of CO_2_ decreases as the temperature increases.

In the model that considers the influence of the surface charge on the pH in the slurry, the pH_L_ in the liquid phase in the slurry increases depending on the amount of charge in the slurry, which, in turn, depends on the surface area and the size of the gel in the slurry. A large *β* is equivalent to large amounts of protonated amine groups that affect the charge balance in the liquid phase in the slurry. As the pH_g_ inside gel cannot be controlled and monitored directly, the presence of the liquid phase in the slurry is important for effectively controlling the pH_g_ for the CO_2_ separation and recovery in this process. As discussed above, pH_g_ in the gel significantly affects the CO_2_ adsorption performance. From the pH dependency of the dissociation curves, an effective operational condition can be proposed based on the dissociation curves shown in Figure 9. When pH_g_ > 8, electrostatic adsorption sites (i.e. the amount of protonated amine in the gels) decrease with the increase in pH_g_ in DMAPAA (p*K*_a_ = 10.35) due to the deprotonation of the amine groups. Thus, pH_g_ < 8 is preferable to keep the amount of electrostatic adsorption sites at maximum. At pH_g_ > 8, the CO_3_^2-^ ions increase due to the pH dependency of the bicarbonate dissociation equilibrium (*β*). This also reduces the total adsorption amount of CO_2_ in the gel due to the charge compensation of the CO_3_^2-^ ions that are twice larger than the HCO_3_^-^ ions in charge. At pH_g_ < 8, the amount of HCO_3_^-^ decreases with the decrease in pH_g_ in the gel. Therefore, for the gel with DMAPAA, the condition with pH_g_ = 8 will achieve the largest adsorption amount from the viewpoint of pH. To desorb CO_2_ in the gel effectively, pH_g_ should be decreased below the p*K*_a_ line to increase *Q*. As shown in this model, the pH in the gel depends on the ion concentration in the solution, the fixed charge concentration and the Donnan ratio. The cyclic capacity of the gel is effectively usable by controlling the solution pH based on these conditions.

In the case of the DAC process, it requires the small regeneration and small desorption energy, a large capacity of the adsorbent for the low concentration CO_2_ in the atmosphere, and the long life of the adsorbent. In this study, low temperature desorption/regeneration (small desorption/regeneration energy) for dilute CO_2_ were achieved. However, the slurry consumes a large amount of heat in the temperature swing of the slurry due to its water content. This is another reason why low-cost heat is a factor in the practical application of this system. The performance of this gel system was evaluated using the developed equilibrium sorption model. A comparison of the CO_2_ recovery performance of the gel system with a similar DAC process [3] is shown in Appendix A as supporting material. The gel system consumes the large amount of heat in the temperature swing process of the slurry, resulting in larger specific energy requirements for 1 kg CO_2_ recovery. This energy consumption needs to be reduced by reducing the water content of the slurry as much as possible. Since the gel does not lose its temperature responsiveness even if the water content is lowered, the development of a gel with CO_2_ adsorption/desorption function even at a low water content will be effective. It is also important to reduce the amount of water entrained with the CO_2_ recovery by precise temperature and moisture management in the adsorption process and pre-treatment of air (fill in advance with water vapour before introducing into reactor, etc.). Sensible heat and latent heat will also be significantly reduced by their recovery using heat exchangers or heat pumps.

This process uses a large amount of the synthesize gels with acrylamide-derived backbone polymers. Thus, it will be necessary to consider the environmental impact of the synthesized gels in future implementations. The cytotoxicity and neurotoxicity of acrylamide derivative monomers are known, but no reports have been found for their polymers [32]. The risk of the use amines would be reduced, because the amine is chemically immobilized in the cross-linked polymer network of the gels.

## 5. Conclusions

The absorption and recovery of dilute CO_2_ were demonstrated using the slurry with the cross-linked thermo-responsive polymer hydrogels immobilizing tertiary amine groups. The influence of hydrophobicity of the polymer network and the amine content on the LCST, desorption behaviour of CO_2_ and cycle capacity for CO_2_ were investigated. The hydrophobisation of the polymer network of the hydrogel enables a large decrease in the temperature of CO_2_ desorption without losing the cycle capacity of the gels, which is induced by the larger decrease of p*K*_a_ of amine in the hydrophobic gels with cooperation of Donnan effect on the pH in the gels. This brings about a reduction in the energy consumed for the desorption of the amine. The pH in the gel is also a key condition for the effective use of the adsorption cycle capacity of the gel when Donnan equilibrium is considered. Since the pH in the gel depends on the number of amine groups and the pH in the slurry liquid phase, the balance of these factors affects the adsorption and desorption of CO_2_ in the gel slurry system; consequently, systematic control of these factors is important for effective CO_2_ separation and recovery. 

The equilibrium adsorption model of CO_2_, which explains the temperature dependence of CO_2_ adsorption/desorption and pH change in the gel slurry system, was constructed considering the Donnan equilibrium and the surface charge contribution of the gels. The model is available to estimate the influence of p*K*a of the immobilized amine groups and solution pH on the CO_2_ recovery performance of the gel slurry system as a function of the temperature of the slurry. 

The recovery of low concentrations of CO_2_ by the gel system and applicability to DAC were examined by the model analysis using the constructed equilibrium adsorption model along with the adsorption data obtained in this study. It is necessary to control the water content of the slurry in order to apply the gel system to DAC. To develop the energy-saving CO_2_ recovery process, a reduction of the water content in the slurry and in the gels is necessary. A further decrease of the specific energy requirement will be possible through (1) evaluation of the lower limit of water content in the slurry, (2) the design of the gels that function with small water contents, (3) heat utilization using a heat exchanger and heat pump, and (4) optimization of the process operation.

## Figures and Tables

**Figure 1 polymers-11-01024-f001:**
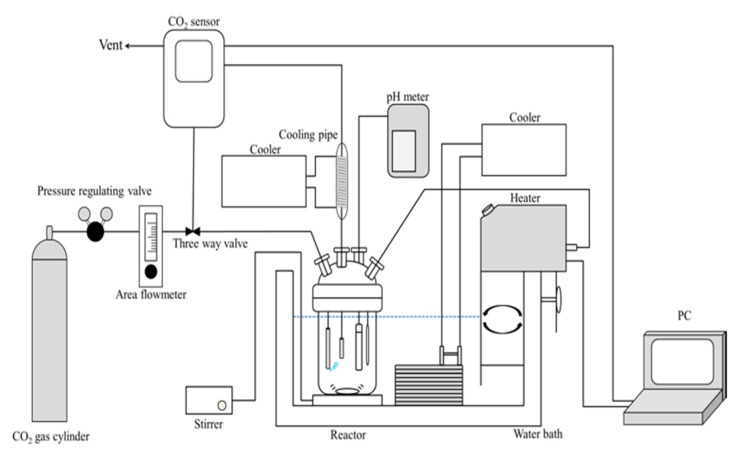
Schematic diagram of the CO_2_ adsorption-desorption experiment.

**Figure 2 polymers-11-01024-f002:**
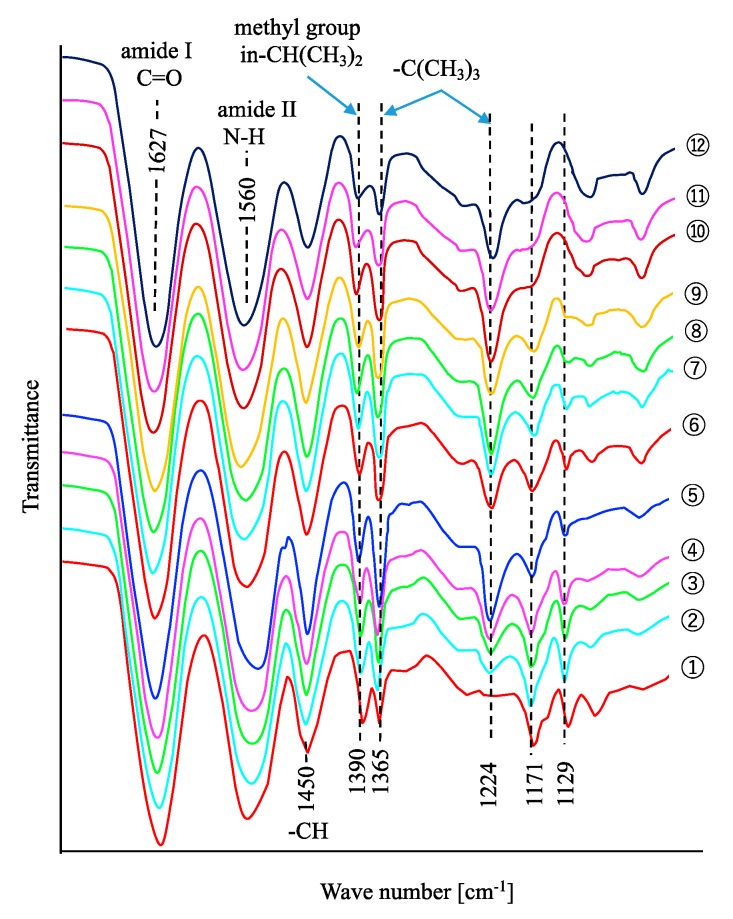
FTIR spectra of the gels.

**Figure 3 polymers-11-01024-f003:**
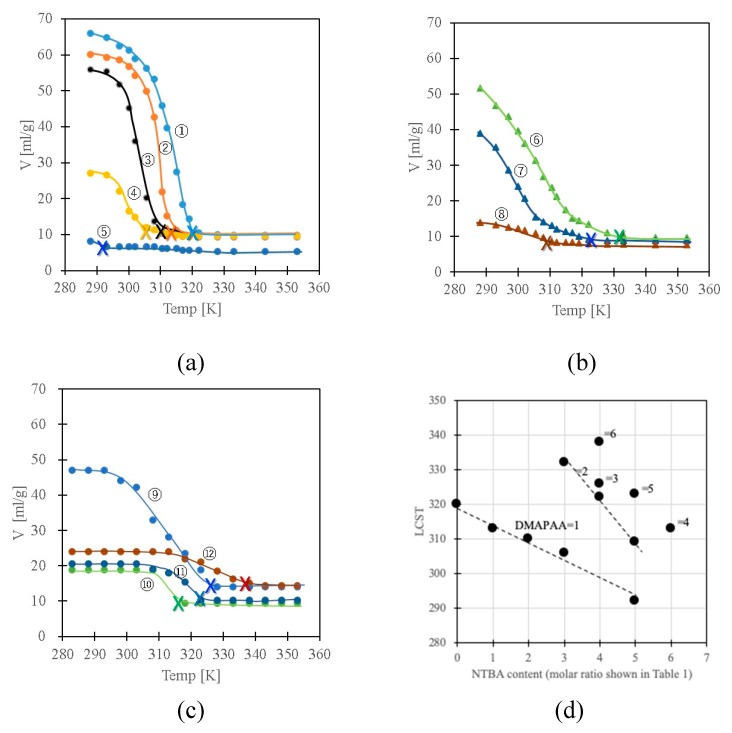
Swelling behavior of the gels (**a**) No. 1–5, (**b**) No. 6–8, (**c**) No. 9–12 and (**d**) LCST plots. The numbers in Figure 3d indicate the molar ratio of DMAPAA shown in Table.

**Figure 4 polymers-11-01024-f004:**
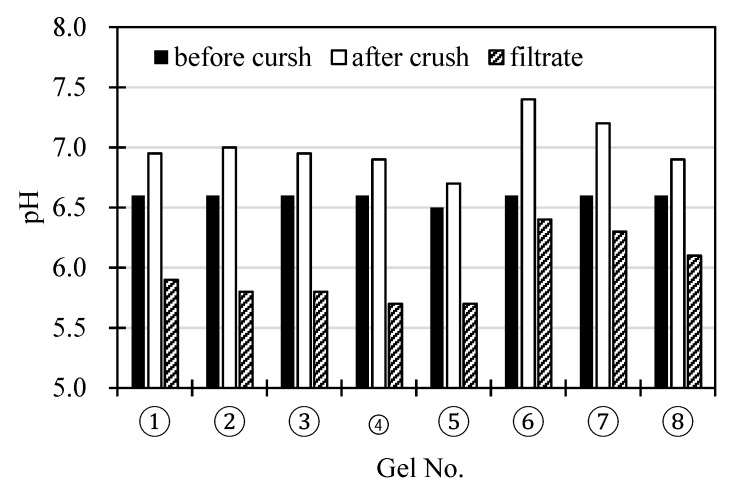
The pH values of the slurries before and after the crushing of the gels in the slurries, and the pH of the filtrates of the slurries that were phase-separated by heating.

**Figure 5 polymers-11-01024-f005:**
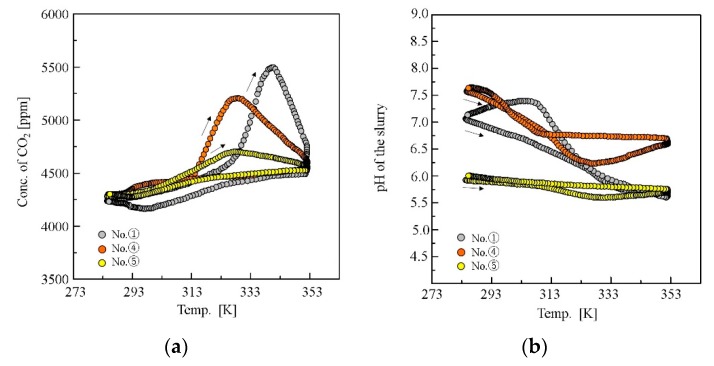
(**a**) Temperature dependence of CO_2_ concentration in the effluent gas from the reactor and (**b**) pH of the slurry during the temperature swing.

**Figure 6 polymers-11-01024-f006:**
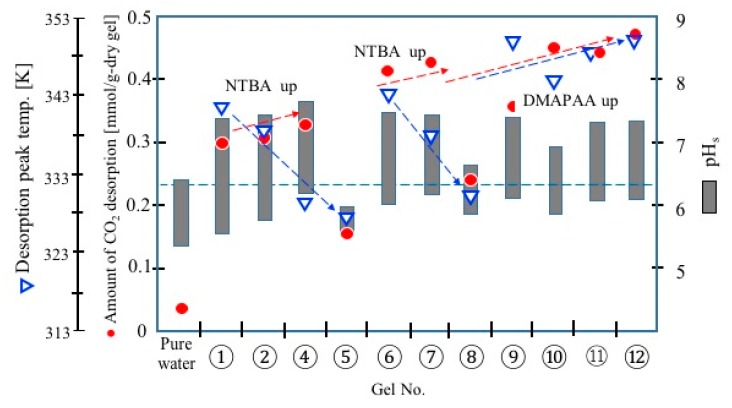
Summary of the CO_2_ adsorption/desorption test.

**Figure 7 polymers-11-01024-f007:**
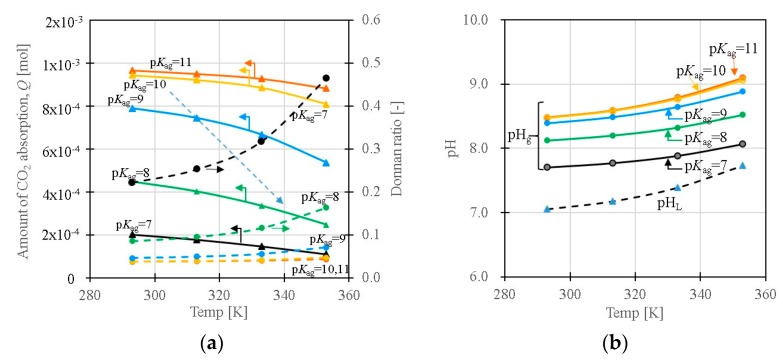
(**a**) Temperature dependence of the amount of CO_2_ absorption; *Q* and Donnan ratio as functions of p*K*_a,g_. (**b**) Temperature dependence of pH_L_ and pH_g_ as functions of p*K*_a_ (*p*CO_2_ = 1000 ppm, [*X*]_0_ = 0.001 mol/L-gel).

**Figure 8 polymers-11-01024-f008:**
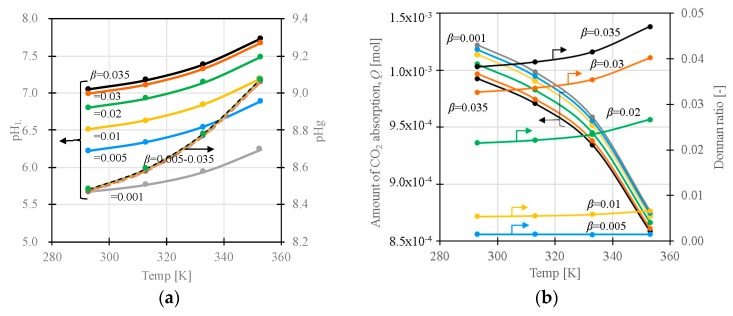
(**a**) Temperature dependence of pH inside and outside the gel as a function of *β*, and (**b**) temperature dependence of *Q* and Donnan ratio as a function of *β* (*p*CO_2_ = 1000 ppm, p*K*_a,g_ = 10, [*X*]_0_ = 0.001 mol/L-gel).

**Figure 9 polymers-11-01024-f009:**
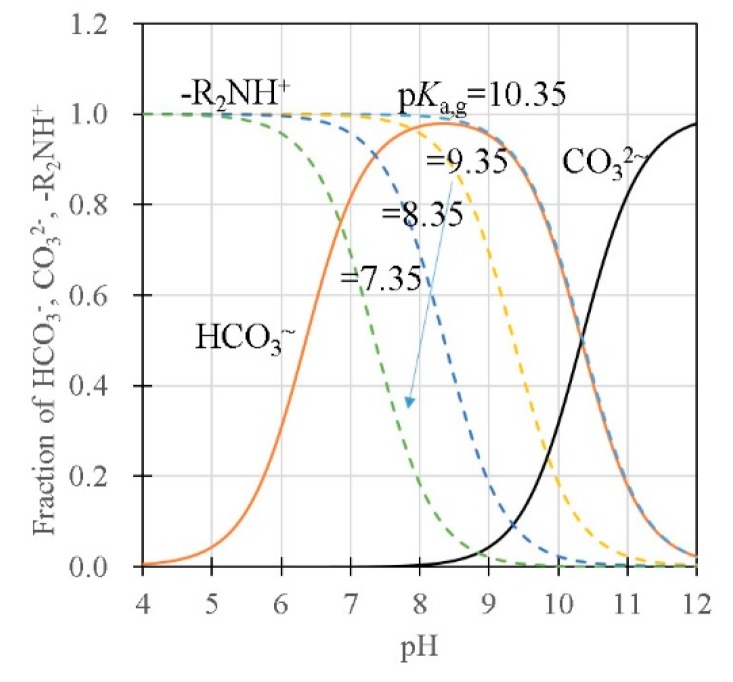
pH dependence of the dissociation for the reaction of bicarbonate dissociation, carbonate dissociation, and dissociation of amine (DMAPAA).

**Table 1 polymers-11-01024-t001:** Composition of the synthesised gels.

Gel No.	Molar Ratio among NIPA, NTBA and DMAPAA (Total 700 mM)			BIS [mM]	mmol of DMAPAA /1 g Monomer Mixture
	NIPA	NTBA	DMAPAA		
①	9	0	1	35	0.80
②	8	1	1	35	0.79
③	7	2	1	35	0.78
④	6	3	1	35	0.77
⑤	4	5	1	35	0.76
⑥	5	3	2	35	1.50
⑦	4	4	2	35	1.48
⑧	3	5	2	35	1.47
⑨	3	4	3	70	2.04
⑩	0	6	4	105	2.47
⑪	0	5	5	105	3.03
⑫	0	4	6	105	3.58

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
