# Peer review of "Influence of Hydrophobicity of Backbone Polymer in Thermo-Responsive Hydrogel with Immobilized Amine on Cycle Capacity for Absorption and Recovery of CO2"

_polymers, 2019, doi:10.3390/polym11061024_

Round 1
Reviewer 1 Report
In this article, authors explored various polymer hydrogels with a lower critical solution temperature (LCST) polymer network to develop a separation and recovery process that is capable of desorbing CO2 at low temperatures, with minimum energy consumption. It also details the adsorption and desorption phenomenon of carbon dioxide through the mathematical modeling. This work is an example of a new application of thermo-sensitive hydrogels with a novelty. If only a few changes are made, they will be published in `Polymers’.
1. Supply LCST of your synthesized gels (no. 1 ~ 12) with x mark of figure 3.
2. It is required that FT-IR spectrum data of gel no. 6 ~ 12 and comparisons among no. 1 ~ 5.
3. A discussion of the economic feasibility of applying to actual chemisorption of synthesized materials. And a brief description of the biological safety of synthesized hydrogels is required in the discussion.
4. Although we describe the adsorption and desorption of carbon dioxide for each condition, there is a lack of explanation as to which composition is most advantageous for real-life applications. For example, it is necessary to discuss the very cold conditions such as Antarctic/Arctic or room temperature with pH 7 or utilization of synthesized hyrogels in the temperate zone where the global population lives the most.
Author Response
I appreciate very much for the review of our manuscript and the valuable comments.
The manuscript was revised based on the reviewers' comments. The modified parts were hatched in yellow. Followings are the answers to each comment.
1. The LCST of the gels were supplied in the main text and plot of the LCST as the function of NTBA content was added as Fig.3(d) for easy understanding of the LCST trend.
2. FT-IR spectrum data of gel no.6~12 were added in Fig.2. The FT-IR spectra of all the synthesized gels were compared to confirm the certainty that the gels with the designed compositions were obtained.
3. Although this process achieved a low temperature in the desorption/regeneration process, a large amount of water with a large heat capacity in the slurry requires a large amount of energy in the temperature swing operation. For this reason, the use of renewable energy or waste heat is the premise in practical application. The electric power of the blower consumed to bring the treating gas into contact with the slurry will also be large in the gas-liquid contact process. To produce the energy-saving CO2recovery process, optimization of the water content and the design of contactor, as well as the design of the gels, are very important. Since the purpose of the present study is to develop the mathematical model of the system with thermo-responsive gel slurry for the systematic evaluation of the process in the next stage of this research, we would like to report the economics of the process in detail in the next paper.
A brief description of the biological safety of the gels was added at the end of the discussion section.
4. Thank you very much for the very valuable comment. We also recognize that the discussions such as the optimum composition of the gel, the optimum operational conditions that achieve energy saving, the requirements on the location of the system establishment and target applications, are very important. For the composition of the gels, the gel with high amine content with a sharp volume phase transition at a low temperature is ideal for the energy saving effective recovery of CO2, but an increase in the amount of NTBA and DMAPAA results in the gels that exhibit a gentle temperature-dependent volume change as indicated in this study. As a result, larger heat energy will be required due to an increase in the swing range of temperature. We consider that it is necessary to optimize the operational conditions as well as the gel composition depending on environmental conditions of where the system will be located, in order to increase the efficiency of the CO2recovery comprehensively. A systematic process evaluation is necessary to find the answers to the discussions (economy, process time, cycle capacity, etc.). We consider comprehensive economics and applicability of the process should be discussed along with the model analysis of the process. We are currently examining process evaluation and will report in the next paper.
We hope the revised manuscript is acceptable for publication.
Reviewer 2 Report
The manuscript reported hydrogels for co2 adsorption and desorption using corresponding instruments. The manuscript is very well-written and the technical quality is excellent. This work should be of interests to the researchers working in the fields of fucntional materials. I will recommend the manuscript could be published.
Author Response
I appreciate very much for the review of our manuscript. The manuscript was revised based on the reviewers' comments. The revised parts were hatched in yellow in the revised manuscript. We hope the revised manuscript is acceptable for publication. Thank you very much again.
Reviewer 3 Report
The authors do describe interesting experiments with amine containing gels varying hydrophobicity and swelling behaviour in water as a function of pH too. In itself the work is described and carried out properly.
However it is not taking into account the basic requirements for Direct Air Capture of CO2, with respect to 1) its low concentration (~400 ppm only) and related to that, 2) the large amount of air required (~1400m3) to capture 1 kg of CO2. Point 1) requires a significant physi/chemisorption energy as otherwise the capacity of the gel will not be used (cost) effectively, and point 2) will lead to a huge amount of H2O that either will condense or evaporate as a consequence of various saturation amounts of H2O in the atmosphere. The latter will then dominate the energy consumption of the DAC process. Not to mention the use of (volatile) alcohols which easily will evaporate.
Author Response
I appreciate very much for the review of our manuscript and the valuable comments.
As the reviewer pointed out, DAC process requires the small regeneration and the desorption energy, the fast adsorption rate, the large capacity of the adsorbent for very low concentration of CO2, and the long life of adsorbent. Besides that, in order to save the electric power consumed by the blower, an efficient contactor with a small pressure drop and minimum capital cost of the facility are needed. Needless to say, it is necessary to optimize the adsorbent design and the operation conditions of the process as a whole from the viewpoint of economy and energy saving. In a gas-liquid contact reactor using the gel slurries, relative humidity inside the reactor will be high and the condensation of water vapor in the treating gas will not occur during the CO2adsorption process unless the temperature of the reactor is lower than the temperature of the environment (the temperature of the air to be treated). On the other hand, in the desorption process, the reactor is heated in a partially closed condition to recover CO2with higher concentration, the energy consumed by the evaporation of water will be very small compared with the energy consumed on the whole process. The operating conditions have to be optimized in the gel slurry system depending on the temperature of the atmospheric environment whatever the process conditions. Of course, it is the premise that the large energy required for the temperature control in the process can be secured by the use of low carbon renewable energy or waste heat. In the temperature-responsive gel slurry system, the matching of temperature operation and thermo-responsive phase behavior of the gels is the very important control factor to obtain an efficient optimum CO2recovery process. The model that represents the process under study is essential to perform a comprehensive assessment of the process performance. In this study, we have experimentally determined the CO2adsorption cycle capacity in the temperature-responsive gel slurry that has hydrophobized networks, and are focusing on developing a basic model of this system based on the phenomenological facts observed in the experiments performed in this study. We consider economics and applicability of the process should be discussed along with the model analysis of the process. Thus, we are currently examining process evaluation and would like to report in the next paper.
 The requirements in the DAC process were added in the introduction part. The advantages and prerequisites in the use of the gel slurry for the CO2recovery process were also indicated clearly in the introduction part. The manuscript was revised based on the reviewers' comments. The purpose of this study was specifically indicated in the revised manuscript. Subjects that will be needed to evaluate the gel slurry process were indicated in the discussion part. Based on the above revision, the conclusion was revised in part. The revised parts are hatched in yellow.
We hope the revised manuscript is acceptable for publication.
Round 2
Reviewer 3 Report
I think the authors did not understand my basic concern, so I apologise for being not more clearer.
I will give it another try: Standard conditions for DAC:
CO2 in the atmosphere: 400 ppm = 0.04%
H2O in the atmosphere: 0 - 4%
Assume 0% H2O present and 2% H2O as saturation level in treated air, than for each kg CO2 captured 2%/0.04%*18/44 = 20,45 kg H2O/kg CO2 will evaporate from the hydrogel (!). Or assume the reverse, and 20.45 kg of H2O will condense for each kg of CO2. Thus atmospheric H2O evaporation/condensation will dominate the energy consumption of the DAC process, on top of the thermodynamically unavoidable amount of energy required to concentrate CO2 from 400-1.000.000 ppm (=pure) CO2..
So the requirements for the gel are: as high as possible reversible CO2 uptake per kg, and an absolute low water uptake/release at relevant process temperature.
What should be demonstrated by the authors is that their proposed system for capturing CO2 (which it does) can avoid large condensation/evaporation from atmospheric H2O by adapting gel- or other process parameters. For example a gel that has an usable capacity 100g of CO2/kg (at 400 ppm )and a saturated H2O content of 10-20 g/kg in the overall system.
The application of (volatile) alcohols in the system is detrimental as they will evaporate in large amount in the air flow anyway!
Still, the chemical work itself has been carried out properly. If this system would not be suited for DAC, it might be useful for some medical application (controlled drug release, or pH/CO2 dependent drug release) in another journal.
Author Response
Dear Sir
Please find the attached file of the answer letter.
Sincerely
Y.Seida
Toyo Univ., Japan

Round 3
Reviewer 3 Report
Dear authors,
I am satisfied with your respones and additions you made in the artcile and Supporting Information.